# Visual Side Effects Linked to Sildenafil Consumption: An Update

**DOI:** 10.3390/biomedicines9030291

**Published:** 2021-03-12

**Authors:** Eva Ausó, Violeta Gómez-Vicente, Gema Esquiva

**Affiliations:** Department of Optics, Pharmacology and Anatomy, University of Alicante, 03690 Alicante, Spain; eva.auso@ua.es (E.A.); vgvicente@ua.es (V.G.-V.)

**Keywords:** phosphodiesterase, guanylyl cyclase, viagra, retinal toxicity

## Abstract

Phosphodiesterase type 5 (PDE5) inhibitors such as Viagra^®^ (sildenafil citrate) have demonstrated efficacy in the treatment of erectile dysfunction (ED) by inducing cyclic guanosine monophosphate (cGMP) elevation followed by vasodilation and increased blood flow. It also exerts minor inhibitory action against PDE6, which is present exclusively in rod and cone photoreceptors. The effects of sildenafil on the visual system have been investigated in a wide variety of clinical and preclinical studies due to the fact that a high dose of sildenafil may cause mild and transient visual symptoms in some patients. A literature review was performed using PubMed, Cochrane Library and Clinical Trials databases from 1990 up to 2020, focusing on the pathophysiology of visual disorders induced by sildenafil. The aim of this review was not only to gather and summarize the information available on sildenafil clinical trials (CTs), but also to spot subpopulations with increased risk of developing undesirable visual side effects. This PDE inhibitor has been associated with transient and reversible ocular side effects, including changes in color vision and light perception, blurred vision, photophobia, conjunctival hyperemia and keratitis, and alterations in the electroretinogram (ERG). Sildenafil may induce a reversible increase in intraocular pressure (IOP) and a few case reports suggest it is involved in the development of nonarteritic ischemic optic neuropathy (NAION). Reversible idiopathic serous macular detachment, central serous retinopathy and ERG disturbances have been related to the significant impact of sildenafil on retinal perfusion. So far, sildenafil does not seem to cause permanent toxic effects on chorioretinal tissue and photoreceptors as long as the therapeutic dose is not exceeded and is taken under a physician’s direction to treat a medical condition. However, the recreational use of sildenafil can lead to harmful side effects, including vision changes.

## 1. Introduction

### 1.1. Phototransduction Cascade

The phototransduction process is a G-protein mediated signaling cascade where rod or cone opsins couple photon absorption to current flow at the photoreceptor outer segment plasma membrane [1]. In the dark, cyclic guanosine monophosphate (cGMP), which is at a relatively high concentration in the photoreceptor outer segment, binds and maintains cyclic nucleotide-gated (CNG) channels in the plasma membrane in an open state, resulting in an influx of Na^+^ and Ca^2+^ ions into the cytosol. To maintain Ca^2+^ homeostasis within the photoreceptor, K^+^ and Ca^2+^ are, in parallel, continuously extruded via the potassium-dependent sodium-calcium exchanger (NCKX). This constant inward current, referred to as the dark current, causes photoreceptor depolarization and glutamate release at the synaptic terminal, inhibiting postsynaptic second-order neurons (bipolar cells). Absorption of photons by rhodopsin leads to the sequential activation of G-protein transducin and phosphodiesterase 6 (PDE6), responsible for the hydrolysis of cGMP and the consequent closure of CNG channels. This interrupts the dark current, resulting in the hyperpolarization of outer segments due to the continued activity of NCKX. As a result, the generation of this electro-chemical signal halts the release of neurotransmitters at the photoreceptor axon terminal and the visual signal is propagated to postsynaptic cells [1].

The role of cGMP as a second messenger is key in the regulation of phototransduction since the whole signaling cascade depends on the balance between its synthesis by retinal guanylyl cyclase (GC) and its hydrolysis by PDE6. Thus, it seems obvious that the disruption of cGMP metabolism implies serious consequences for visual functioning, including photoreceptor toxicity and cell death (reviewed in [2]). Processes such as retinal oxidative stress entail the generation of reactive nitrogen intermediates such as nitric oxide (NO), a second messenger that stimulates retinal GC, increasing free cGMP levels [3]. Likewise, genetic mutations are also involved in the pathological intracellular concentrations of cGMP. For instance, some forms of inherited retinal degeneration such as retinitis pigmentosa, Leber congenital amaurosis, or cone–rod dystrophies are related to increased cGMP levels (reviewed in [4]). Moreover, pharmacologically targeting the cGMP pathway has been postulated as a novel and interesting therapeutic approach for the treatment of inherited retinal degenerations [5]. Although the mechanisms linking elevated cGMP to photoreceptor demise have not been completely elucidated yet, two targets of cGMP, whose overactivation contributes to rod cell death, have been proposed: protein kinase G (PKG) and CNG channels [6].

It is known that the NO/GC/cGMP/PKG signaling pathway is functional and widely distributed in specific cell types of both the internal and external retina of mice [3]. Studies performed in the murine models of retinitis pigmentosa *rd1* and *rd10*, which carry loss-of-function mutations in the beta subunit of rod PDE6 [7], have shown that high cGMP levels during retinal degeneration trigger an overactivation of PKG, which contributes to photoreceptor death [8]. Although cGMP-dependent phosphorylation of PKG in photoreceptors has already been demonstrated in 1977 [9], Paquet-Durand’s study was the first to link excessive PKG activity directly to cell death [8]. On the other hand, as mentioned above, cGMP regulates the opening of the CNG channels present in the plasma membrane of the photoreceptor outer segment. Therefore, excessive cGMP alters Ca^2+^ homeostasis, impairing the function of Ca^2+^-dependent phototransduction proteins such as recoverin and guanylyl cyclase-activating proteins (GCAPs), and even triggering photoreceptor death via the calpain signaling pathway [1,10] (Figure 1).

### 1.2. Phosphodiesterases and Inhibitors

Phosphodiesterases are a family of enzymes that regulate intracellular levels of the second messengers cAMP and cGMP. Although phosphodiesterases are found in every cell in the body, the distribution of isoenzymes varies between tissues. For instance, PDE5 is expressed in vascular smooth muscles (prominently expressed in the penis corpus cavernosum), skeletal muscles, and many other tissues including kidney, pancreas, heart, lung, liver, brain, placenta and various gastrointestinal tissues [11]. By contrast, PDE6 is present exclusively in retinal photoreceptors [12]. The PDE6 family consists of three genes (PDE6A, PDE6B and PDE6C) that encode three catalytic subunits (α, β and α′, respectively). The α and β subunits are expressed predominantly in rods, whereas the α′ subunit is expressed in cones [13]. cGMP binds to PDE6 through two GAF domains (GAF-A and GAF-B) at the amino-terminal end of the enzyme. The structural similarity of PDE isoenzymes catalytic domains results in poor specificity of inhibitory drugs. In this sense, it is worth mentioning that first-generation PDE5 inhibitors such as sildenafil (Viagra^®^), vardenafil (Levitra^®^), and tadalafil (Cialis^®^) are highly selective for PDE5 and represent the first successful application of PDE inhibition therapy to an individual isoenzyme. Nonetheless, despite this high selectivity, each of these drugs inhibits other PDE isoenzymes to some extent. Sildenafil and vardenafil, for example, have shown only 10 and 15 times lower specificity for PDE6 than for PDE5, respectively (reviewed in [14]), which could be explained by the fact that the kinetic and catalytic properties of PDE6 are very similar to those of PDE5 [15,16,17].

PDE5 inhibitors have been used for the treatment of erectile dysfunction (ED), which is a form of peripheral vascular disease that impairs men’s abilities to achieve and maintain an erection, and have become some of the best-selling medications worldwide. Sildenafil and its analogues operate by increasing the cGMP levels because they occupy the active site of PDE5 and prevent cGMP catalysis. cGMP acts as a powerful smooth muscle relaxant that promotes blood flow to the corpus cavernosum, facilitating penis erection [18]. Apart from their use as drugs to treat ED, the US Food and Drug Administration also approved the use of sildenafil and their analogues for the treatment of pulmonary arterial hypertension (PAH) in 2005. These inhibitors offer the possibility of improving the patient’s quality of life, as well as being candidate drugs for palliative therapy [19]. In addition to the three drugs mentioned above, the second-generation inhibitor avanafil (Stendra^®^) became internationally available in 2013. Avanafil exhibited 100 times lower specificity for PDE6 than for PDE5, presumably reducing the potential side effects derived from the nonselective inhibition of PDE6 by sildenafil and vardenafil (reviewed in [14]). Other second-generation (udenafil and mirodenafil) or third-generation (lodenafil, SLX-2101, JNJ-10280205, and JNJ-10287069) PDE5 inhibitors have been either approved and introduced into the market in some parts of the world or are at the final stages of their clinical development. Udenafil (Zydena) is only available in some Asian countries and Russia, mirodenafil (Mvix) is commercialized in various Asian countries and lodenafil (Helleva) is sold in Brazil [20]. All of them have been trialed in tablet formulations at different doses, whose broadest range spans from 25 to 200 mg [21]. Several studies indicate that, in general terms, PDE5 inhibitors are well tolerated and their side effects are few, mild and very similar among the different compounds studied, except for tadalafil, which caused a higher incidence of myalgia (Table 1). Many of the side effects are due to the vasoactivity of these compounds, given the expression of PDE5 in vascular smooth muscles. The most common reported dose-dependent adverse events include headache, flushing, nasal congestion, facial and ocular hyperemia, myalgia, back pain and dyspepsia [22,23,24].

The occurrence of side effects increases with both serum levels and exposure time to the PDE5 inhibitor [25]. To overcome these issues, novel drug formulations that improve the safety and efficacy profile of the drug are under development. Despite the side effects, oral administration of PDE5 inhibitors (tablets, oral solution or orodispersible tablets) is nowadays considered the first-line therapy for ED. A second-line treatment consists of invasive procedures such as intracavernosal injections with vasogenic drugs such as alprostadil (synthetic prostaglandin E1), papaverine or phentolamine, as well as intraurethral alprostadil suppositories and vacuum erection devices. These show a more favorable systemic side effect profile compared to oral pharmacotherapy [26] and, despite being invasive, they are widely used. To avoid invasive techniques and, at the same time, minimize systemic side effects, topical formulations (alprostadil and sildenafil topical cream) constitute a promising alternative, as they can be applied locally and are safe and easy to use [21,27,28]. Additionally, solid lipid nanoparticles in hydrogel films for the transdermal local delivery of avanafil have been assayed in vitro and ex vivo with success [29]. Moreover, emerging medications and procedures are currently under investigation for the treatment of ED in both preclinical and clinical settings, including non-PDE5 inhibitor oral drugs such as melanocortin receptor antagonists (subcutaneous melanocortin analogue (PT-141)), rho-kinase inhibitors (SAR407899), and soluble GC activator (BAY60-4552 and BAY 60-2770) [21,27,28]. Additionally under consideration are: regeneration therapy involving stem cell injection; gene therapy where the genetic material can be easily injected into the penis; low-intensity extracorporeal shock wave therapy; low-intensity pulse ultrasound; platelet-rich plasma injections [21,28,30]. Finally, the use of nanotechnology for drug delivery is being studied in murine models for all delivery methods (oral, topical, and intracavernosal) as a way to either enhance bioavailability or to improve and promote the local effects of medications [6].

### 1.3. Side Effects of Sildenafil

Of the above mentioned drugs, sildenafil is the one that has exhibited a higher incidence of visual side effects, given that it is only 10-fold more potent on PDE5 than on PDE6 [11,31]. As an example, numerous case reports describing ocular side effects associated with the consumption of sildenafil can be found in the medical literature (Table 2) [32,33,34,35,36,37,38,39,40,41,42,43,44,45,46,47,48,49,50]. Its nature of use, its frequent use, a possible overdosage beyond the recommended optimal dose and the advanced age of the patient with frequent associated vascular pathologies makes it necessary to mention some of the adverse effects observed by the intake of sildenafil. Sildenafil administered orally is rapidly absorbed and maximum plasma concentrations occur within 30–120 min. In Spain, the Centre for Information on Medicines reports that the recommended dose for patients with PAH is 20 mg three times a day. In contrast, for the treatment of ED, it should only be used before sexual relations, with an optimum single dose of 50 mg once a day for adults. The dose can be reduced to 25 mg or increased to 100 mg a day (maximum single dose) depending on individual tolerance and efficacy. However, the consumer could alter this prescription by taking a dose above the recommended level to achieve good results. Because of this, there is a need for control since an intake of 100 mg increases its toxicity five-fold [38]. Additionally, sildenafil pharmacokinetics can be modified by the concomitant use of other drugs such as inhibitors of the cytochrome P450 (CYP) 3A4 (e.g., macrolide antibiotics, calcium channel blockers, etc.) [51], which is the main enzyme responsible for its hepatic metabolization. Inhibition of CYP3A4 would elevate the plasma concentration of sildenafil, thereby also increasing the likelihood of unwanted side effects. These key drug-metabolizing enzymes often display genetic polymorphisms that contribute to the individual variability in drug safety and efficacy among patients and may represent a risk of drug–drug interactions [52].

Overall, the most common adverse effects of sildenafil are strongly associated with its pharmacological nature as an inhibitor of PDE5 (headache, nasal congestion, ageing and dyspepsia) and as a weak inhibitor of PDE6 (visual impairment), being dose-dependent and observed in 6–18% of men taking sildenafil [53]. In this sense, visual side effects were reported in 3–11% of men taking 25–100 mg of sildenafil, 50% of men taking 200 mg and 100% of men taking 600 or 800 mg [31,54,55,56] (center for drug evaluation). Although subjective visual changes are common, studies on healthy volunteers [55,57], men with ED [54,58] and patients with previous visual pathologies such as age-related macular degeneration (AMD) [56] who were taking sildenafil either as a single dose [55,56,57] or following a long-term treatment [54,58] have not found significant differences in psychophysical testing of visual function, except for color discrimination, predominantly in the blue–green range, in some studies [59]. The effects on retinal function are shown as modest and transient visual symptoms, commonly reported as blue vision, increased sensitivity to lights and blurred vision, more often at high doses [41,60]. Karaarslan’s study has reported visual symptoms up to 21 days after taking sildenafil [41]. Although the cause of blue-tinted vision is unknown, it is thought that it can be related to PDE6 inhibition in the retina [61] but data are nonconclusive [62]. Because PDE5 is expressed in the endothelial and smooth muscle cells of the choroidal and retinal vessels, sildenafil may affect ocular blood flow [63]; thus, it is reasonable to think that may cause other visual symptoms apart from those derived from the nonselective inhibition of PDE6 [11,64]. In fact, severe effects such as an increase in intraocular pressure (IOP) [65,66,67], retinal and choroidal vasodilation and altered blood flow [68,69], and nonarteritic anterior ischemic optic neuropathy (NAION) [45,70,71] have been reported as a consequence of the intake of sildenafil. Since many of the symptoms are dose-dependent, further studies are needed to establish the dose above which adverse effects occur in sildenafil users.

The purpose of this literature review was to gather and summarize the information available on sildenafil clinical trials (CTs), focusing on the possible adverse effects related to different aspects of visual health.

## 2. Results

Given that sildenafil nonselective inhibition of retinal PDE6 results in visual disturbances, several reports have questioned the ocular safety of this drug over the last two decades [11,53,64,72,73]. Accordingly, several CTs have been conducted to evaluate the incidence of sildenafil-associated visual side effects, as well as its safety. Specifically, between the years 1999 and 2020, in the Cochrane Library (www.cochranelibrary.com, accessed on 20 December 2020) we retrieved 2001 entries including the term “sildenafil” in the title, abstract or keywords, of which we curated contents and selected all those results related with diverse features of the eye’s anatomy and physiology. The first CT on this topic “The effects of sildenafil citrate (Viagra^®^) on color discrimination in volunteers and patients with erectile dysfunction” (CN-00675062) was published in 1999 and over the following decade (2000–2009) a total of twenty-six CTs were conducted to assess adverse effects of sildenafil regarding visual health. Later on, the number of CTs declined and in the 2010–2019 decade only seven trials were registered (Figure 2a and Table 3). Studies carried out on healthy volunteers comprised most of the clinical trials (50%), although studies on ED or AMD patients were also widely represented (approximately 15% each). The remaining 20% is divided between PAH (10%), chronic open-angle glaucoma (COAG) (5%), and ischemic stroke (IS) (5%) patients (Figure 2b and Table 3).

Of the thirty-five registered CTs found on this subject in the US National Library of Medicine databases PubMed (www.pubmed.ncbi.nlm.nih.gov, accessed on 20 December 2020) and ClinicalTrials (www.clinicaltrials.gov, accessed on 20 December 2020), and in the Cochrane Library, twenty-six performed ophthalmic examinations, of which seven assessed ocular anatomy (external inspection of the eye, slit-lamp and fundoscopy), thirteen evaluated IOP, and six measured blood flow. Regarding visual function and perception, eleven CTs evaluated visual acuity (Snellen test), seven assessed contrast sensitivity (Pelli–Robson test), seven measured static perimetry (Humphrey visual field test), four performed electroretinogram (ERG), and nine assayed color perception. In some of the trials (five), the most common transient and subjective visual adverse events, derived from sildenafil consumption, were also evaluated. These included color perception distortion (blue color tinge, color discrimination alterations and tritanopia), changes in sensitivity to light, and blurred vision. In two of the CTs (NCT00461565, NCT01830790), the main outcomes were not publicly available (Table 3).

In general, long-term CTs mainly evaluate the effects of chronic sildenafil uptake on retinal function in patients with a previous pathology such as ED, early-stage AMD, or PAH. By contrast, studies that assessed the effects of a single acute dose of sildenafil were mostly carried out in healthy individuals. The frequency of ED increases dramatically with age and in the presence of cardiovascular risk factors. However, quantitative studies of side effects in both healthy volunteers and patients have produced mixed results.

### 2.1. Ophthalmologic Examination

#### 2.1.1. Ocular Anatomy

Seven studies examined ocular anatomy (anterior and posterior chambers, lens and fundus) using a slit-lamp and/or a fundoscope (or ophthalmoscope). Most CTs were carried out on patients with previous pathologies such as ED [54,58,87,91], with the exception of the studies of Dündar et al. [57] and Yajima et al. [75], which were conducted on healthy volunteers. No study revealed clinically important and significant differences compared with placebo or baseline. Nevertheless, novel studies that question these results are emerging. In a case-control report, which included patients with PAH under chronic sildenafil medication and patients not taking the drug, 33% of the medicated patients showed severe bilateral keratitis. Connective tissue abnormalities are often present in PAH patients but this condition might be exacerbated with the use of sildenafil [69]. In a recent retrospective case report, carried out on seventeen men with ED taking sildenafil for the first time, pupil diameter assessment revealed symmetrical pupillary dilation for all patients, although approximately half of the subjects exhibited abnormally dilated pupils [41]. These current findings make it advisable to refer patients with chronic pathologies who are about to undergo sildenafil treatment for routine ophthalmic assessment with emphasis on the ocular surface evaluation.

#### 2.1.2. Intraocular Pressure

Sildenafil inhibition of PDE5 leads to an increase in the concentration of cGMP, which, in animals, is known to lower IOP [98]. However, no clinically important acute adverse effects of sildenafil on IOP have been reported after 50–150 mg administration on a single [58,75,85] or on two separate doses [77,81,99]. Metelitsina et al. examined the foveal choroidal blood flow in men with AMD and, despite finding a decrease in blood pressure after 30 min of a single sildenafil dose, there were no relevant changes in IOP [86]. Regarding long-term studies, Dündar et al. assessed IOP after 3 months of sildenafil regular use in a group of men with ED and no significant changes were found in this case neither [54]. Likewise, no evident effect on IOP was found after chronic sildenafil administration over 2–4 years for the treatment of ED [91] or PAH [94]. On the other hand, Gerometta et al. studied the effect of 100 mg of sildenafil uptake as a single dose and detected an acute transient IOP increase 60 min later [64]. A possible explanation is that this transient rise could be due to an increase in choroidal volume induced by PDE5 inhibition (as mentioned above, PDE5 is expressed in endothelial cells of choroidal vessels) and might be of importance for patients chronically treated with sildenafil, especially glaucoma patients or individuals at high risk of developing the disease. Finally, Grunwald et al. studied the effect of sildenafil at the maximum therapeutic dose of 100 mg in patients with COAG and did not observe statistical nor clinical significant acute alterations in IOP, similarly to findings in placebo control subjects [78]. Therefore, most published reports have not demonstrated an association between sildenafil administration and IOP elevation, considering transient IOP elevations as coincidental.

#### 2.1.3. Ocular Blood Flow

Sildenafil has a strong systemic vasodilating effect and it is known to decrease systemic blood pressure, which could lead to a decrease in choroidal blood flow [100]. However, since the choroid is a vascular tissue, similar to the corpus cavernosum, sildenafil could also have a strong vasodilatory effect, resulting in increased choroidal [101] and ciliary body perfusion via an increment in the posterior ciliary artery flow as a result of vascular smooth muscle relaxation [102]. The ophthalmic artery is the most responsive ocular artery after PDE5 inhibitors administration. Foresta et al. studied the acute effects of 100 mg sildenafil on the ophthalmic artery blood flow velocity and showed that sildenafil increased the flux in a time-dependent manner [63]. Dündar et al. evaluated the effect of a 50 mg dose of sildenafil in retrobulbar hemodynamics. They observed a significant increase in ophthalmic artery peak systolic velocity and in end-diastolic velocity that could be interpreted as an increase in volumetric blood flow [57]. Additionally, Sponsel et al. reported a significant increase in pulsatile choroidal blood flow 110 min after administration of a unique 50 mg dose of sildenafil [103], though the authors did not detect changes in retinal blood flow, neither in the central retinal nor in the temporal short posterior ciliary artery, in accordance with other studies [77,88,104]. Using laser Doppler flowmetry, Grunwald et al. did not find any significant changes in optic nerve head or foveolar choroidal blood flow neither 1 nor 5 h after sildenafil intake [77]. Therefore, although central retinal artery velocities were not changed, dilations of intraocular vasculature resulted in an increase in the mean ocular blood flow after sildenafil intake [84].

Most studies suggest an increase in choroidal blood flow velocity, with a lesser effect on the retinal vasculature in healthy subjects. This may be due to the production of NO, which plays a key role in the local regulation of ocular blood flow, and probably as a consequence of PDE5 inhibition in smooth muscle cells in a time-dependent manner [85]. It is worth noting that Dündar et al. also assessed the effects of sildenafil on ocular hemodynamics of healthy subjects in the long-term and reported no significant changes with chronic use, reflecting a mere temporary vasodilator effect without altering the orbital vasculature [54,87]. Nevertheless, these effects might have clinical consequences and may constitute a risk for ocular ischemia in patients with previous choroidal circulation problems, as is the case of central serous chorioretinopathy or AMD. In fact, there are reports regarding the development of unilateral and bilateral chorioretinopathy upon the use of sildenafil that resolved spontaneously [105]. PDE5 inhibitors are classified as only a possible risk factor for the development of central serous chorioretinopathy though [105], given that no strong evidence has been found in any of the studies performed to date [106]. In AMD, degeneration of the choroid and choroidal microcirculation (choriocapillaris) occurs with age [107] and choroidal blood flow is decreased [108]. As mentioned previously, choroidal blood flow and thickness may increase in response to sildenafil intake, thus sildenafil treatment is suggested as a means of increasing choroidal perfusion so that some CT-treated AMD patients with systemic sildenafil [97]. Birch et al. examined the acute effect of sildenafil administration in patients with early-stage AMD and observed no significantly or clinically relevant changes in visual function [56]. Furthermore, individuals taking sildenafil showed similar vasodilatation values of major retinal veins as the placebo group [88] and no statistically significant changes were detected in the foveolar choroidal circulation of AMD patients [86]. Finally, Coleman et al. evaluated the effect of sildenafil over 2 years in patients with macular degeneration or macular dystrophy and observed maintenance or even an improvement in the photoreceptor layer, concluding that sildenafil is a safe treatment for AMD that offers significant potential for vision retention and recovery [97]. Additionally, several case reports of NAION have been described in patients receiving sildenafil [109] but the relationship of ischemia to drug intake is not clear.

### 2.2. Visual Function and Perception

#### 2.2.1. Visual Acuity

Eleven CTs evaluated the best-corrected visual acuity (BCVA) following sildenafil administration in healthy people [56,57,76,91], AMD [86,90,97], ED [54,58,87,91] and PAH patients [93,94]. The main finding of both acute trials and open-label extension studies was that chronic oral sildenafil treatment did not seem to result in any significant loss of visual acuity. Furthermore, Coleman et al. observed that participants with best vitelliform eruptive macular degeneration remained not only visually stable but a significant improvement of BCVA was also reported [97]. These findings are in agreement with reports in the literature that describe no significant clinical changes in several test scores such as visual acuity and color vision in PAH patients under a chronic sildenafil routine [69]. Conversely, we have found a recent case report of a 37-year-old woman with a history of primary PAH and a 5-year history of oral sildenafil intake who developed outer macular atrophy and exhibited a severe reduction in visual acuity in her left eye [50]. This research is the first one showing an association between long-term use of sildenafil and severe ocular side effects. Thus, it is necessary to warn about the chronic use of sildenafil and its potential risk of adverse visual outcomes.

#### 2.2.2. Color Vision or Discrimination

Nine CTs evaluated color discrimination using different tests. Five studies were carried out on healthy volunteers [55,59,62,79,91], two in AMD [56,90] and three in ED patients [58,59,91]. Most CTs showed no alterations in color vision after taking sildenafil. Supporting these findings, in a recent study where the color vision was assessed upon chronic use of sildenafil in PAH patients, no significant effects were found [69]. However, the remaining CTs failed to confirm these outcomes. In a double-blind placebo-controlled trial, possible acute effects in color discrimination from 1 to 36 h after taking sildenafil (50–200 mg) were assessed in 16 healthy volunteers. Color perception was measured with Farnsworth–Munsell 100-Hue test (FM-100-Hue). A statistically significant and transient increase in FM 100-Hue error scores was noted at 1 h (100 and 200 mg doses) and 2 h (200 mg) after the consumption of sildenafil. Impaired blue–green color discrimination (induced tritanopia) was detected and the error scores correlated with plasma sildenafil concentration [59]. Similar results were found in a case-control unmasked study, where color perception was measured with the Lanthony desaturated Panel D-15 test. Compared with controls, a higher percentage of the subjects who were taking sildenafil committed more errors from the baseline to the 1-hour testing session. This difference was slightly and statistically significant [79]. Recently, it was reported the case of a 57-year-old man who, upon taking a single 100 mg dose of sildenafil for radical prostatectomy, experienced a sensation of unusual brightness of incoming visual stimulation combined with abnormal color vision that persisted beyond 5 h. These effects fully resolved within 7 days after discontinuing sildenafil [37]. Similarly, in a recent case-series study with 17 ED participants who took a single 100 mg sildenafil oral dose for the first time, visual color disturbances were reported by more than 75% of subjects. They did not reach the 90-point normal threshold in color vision assessment, where 5 of these patients had at least one score indicative of a definite impairment [41]. Such cases suggest that a relatively small subpopulation of people are at risk of disturbingly intense side effects upon intake of PDE inhibitors, which supports the practice of starting on a modest dose when prescribed this kind of drugs.

#### 2.2.3. Contrast Sensitivity

Seven CTs evaluated contrast sensitivity in healthy people [62,80,82] and AMD [86,97], ED [58], and PAH patients [94] after sildenafil administration. Two of those studies revealed changes in contrast sensitivity. In a randomized, double-blind, placebo-controlled trial, using a monitor-based color vision test based on a luminance noise technique, cones with different wavelengths were selectively stimulated. Despite the fact that very small, nonsignificant threshold differences from predose baselines were found between the sildenafil and placebo groups for all three cone types, a statistically significant increase in sensitivity was observed during transient tritanopia, which correlated with sildenafil’s peak plasma concentration [62]. Similarly, in a randomized, double-masked placebo-controlled trial carried out in four healthy individuals who took a single oral dose of 50 mg of sildenafil, an 80% increase in contrast sensitivity compared with baseline was reported [80]. These findings are in the same direction as Karaaslan’s study, where 35% of individuals experienced a transient contrast sensitivity impairment and one individual experienced disability that spontaneously disappeared within 5 days [41]. No statistically or clinically significant differences were reported in the other CTs [58,82,86,94,97]. It is worth noting about Coleman’s study that, despite no significant changes in the Pelli–Robson chart test, all participants self-reported improvements in contrast sensitivity and the best vitelliform eruptive macular degeneration patients could see the chart with both eyes. However, the positive effects found in the studies were fully reversible within a few days.

#### 2.2.4. Humphrey Perimetry Test

The 30-2 program on the Humphrey visual field (HVF) analyzer was used to assess the visual field in seven CTs. Three of them were carried out in healthy volunteers [55,74,91], one in AMD [56], one in PAH [94] and two in ED patients [87,91]. In a randomized prospective case-control study, HVF was carried out in the right and left eyes of eight healthy volunteers with both white-on-white (W/W) and blue-on-yellow (B/Y) protocols. One individual who experienced systemic side effects (headache, dizziness, nausea) also developed quadrantanopic field defects, more pronounced on B/Y than on W/W, between 1 and 2 h after taking a single dose of sildenafil [55,74]. No significant or clinical changes were reported in the rest of the CTs. These favorable results suggest that acute sildenafil administration, as well as long-term intake, is not associated with a compromised visual function such as the static perimetry assessment. Despite this, it is clear that more acute and long-term studies are desirable to investigate possible functional/structural effects of this drug both in healthy people and people with different pathologies.

#### 2.2.5. Visual Disturbances: Light Sensitivity, Blurred Vision and Blue Color Tinge

Five CTs evaluated the presence or absence of the most common visual adverse effects such as light sensitivity changes, blurred vision and blue-tinted vision in healthy subjects [62,79], in AMD [56], ED [58] and PAH patients [94]. Luu et al. reported that 8 out of 14 healthy volunteers who received a single oral dose of 200 mg of sildenafil experienced varied subjective visual disturbances, while the placebo control subjects reported no visual disturbances [79]. The visual adverse effects frequently occurred within 1 to 2 h after drug consumption, highlighting an increased light sensitivity (in 5 out of 14 individuals). Other adverse effects included blue-tinted vision, red and blue speckled vision and blurred vision, which were reported by only one subject each. All these visual transient disturbances appeared to be dose-dependent. In the same direction, a double-masked, open-label trial, described mild to moderate-severe adverse transient events, although the incidence was low in all participants with the exception of one individual (in the sildenafil 80 mg group) who developed severe photophobia 72 days after the start of the study [94]. Slight incidence (<7%) of chromatopsia, cyanopsia, photophobia, and visual disturbances after administration of 80 mg of sildenafil three times daily were reported. It is well known that, generally, these visual disturbances resolve within 5 h. A recent case-series study with 17 ED patients reported visual disturbances that persisted more than 24 h in response to a single 100 mg dose of sildenafil [41]. More than 50% of participants exhibited some degree of clinical photophobia, including one severe and one very severe presentation. A high dose of sildenafil (maximum recommended therapeutic dose for ED) may be the cause for the extended durations of visual secondary effects.

#### 2.2.6. Scotopic and Photic (ERG) Responses

Sildenafil intake can be expected to inhibit the phototransduction process, thus inducing changes in the ERG. However, only minimal changes were observed at sildenafil doses ranging from 50 to 200 mg compared with placebo [110]. Indeed, the assessment of the effect of 100 mg of sildenafil in healthy subjects [111] and in ED patients [112] 1 h after oral intake showed a reversible transient decrease in the amplitude of the a- and b-waves (rod-driven). In the ED plus sildenafil group, the treatment also increased full-field ERG implicit times of the scotopic b-wave that were not considered clinically significant [112]. No significant changes in implicit times were observed in healthy subjects [111]. Conversely, other studies reported transient, modest, dose-related increased photopic, but not scotopic, implicit times with a cone function slightly depressed in the macula and the periphery in healthy individuals receiving a single 100 or 200 mg dose of sildenafil [79,113]. Jägle et al. suggested from their ERG results that, when receiving a single 100 mg dose of sildenafil, inner retinal function was affected and prolonged implicit times of rods and cones showed no significant differences, whereas rod responses 1 h after sildenafil intake were raised too [62], contrary to previous reports about significant, transient reductions in the maximum response amplitudes of a- and b-waves [111,114]. Similarly, other studies supported a higher rod sensitivity and a higher rod response to light stimuli, as recorded by ERG 1 and 2 h after the intake of 50 or 100 mg of sildenafil. These findings are consistent with the weak PDE6 inhibition induced by sildenafil [114]. However, all these acute effects on the ERG are not clinically significant in terms of altered light sensitivity or visual function. Cordell et al. studied chronic PDE5 inhibition over 3 to 6 months on a daily basis of 50 mg of sildenafil and demonstrated no evidence of increased implicit times or decreased ERG amplitudes [91]. Furthermore, Zoumalan et al. studied the chronic daily use of sildenafil at higher doses (120–300 mg) for 1–4 years and did not find any toxic effect, only a modest lengthening of cone implicit time that seemed to be restored a few hours later, indicating that any possible retinal toxicity or visual disturbances of sildenafil may be reversible in the short term [115]. These inconsistencies in the effects in ERG recordings may be due, in part, to the use of different doses and group heterogeneity. Taken together, the ERG results suggest that sildenafil doses of 25 or 50 mg entail minimal visual side effects, and at maximum therapeutic doses, sildenafil can cause acute and transient changes in rod and/or cone function without a practical effect on visual performance.

## 3. Discussion

A growing body of evidence points to cGMP as one of the main players in inherited retinal diseases and oxidative stress-induced retinal degeneration. Therefore, it seems reasonable to think that the disruption of retinal cGMP concentration and subsequent Ca^2+^ homeostasis can be detrimental to photoreceptor survival. PDE5 inhibitors such as sildenafil are often used for the treatment of ED and PAH. Although sildenafil exhibits a high selectivity for PDE5, in high doses it is also capable of binding and inhibiting PDE6 nonselectively. Inhibition of both isoenzymes, PDE5 and PDE6, is the main cause of sildenafil visual side effects. PDE5 is expressed in some ocular tissues such as the endothelial cells of retinal and choroidal vessels. PDE6 is exclusively expressed in photoreceptors and its inhibition directly alters the phototransduction cascade due to an increase in cGMP levels. This idea has prompted many scientists and researchers to conduct CT to evaluate the safety and the visual side effects of sildenafil. From 1999, the moment when the first CT assessing visual parameters after sildenafil uptake was published, many other studies have been released and, although visual disturbances have been extensively reported, all of them seem to be transient and mild. However, many case reports regarding ocular side effects linked to sildenafil consumption have recently arisen in the medical literature (see Table 2) [34,37,38,40,41,61,106,116]. It seems that a small subset of individuals experience more severe effects either with a low dose but chronic use of sildenafil (as for the treatment of PAH) or with a high dose but sporadic use of the drug (as for the treatment of ED). Among the different factors that could influence individual sensitivity to sildenafil are gene polymorphisms of CYP3A4 and CYP2C9, the two major sildenafil-matabolizing hepatic enzymes [53]. This draws attention to the necessity of designing and conducting novel CTs where other populations are also represented. For instance, the majority of the CTs were carried out in small group samples and preferentially in males (see Figure 2b). It is obvious that ED affects only males; however, PAH or AMD affect both males and females and, therefore, it is interesting and necessary that both genders are equally represented in the CTs.

Additionally, evidence from preclinical studies carried out in animal models of human retinal diseases suggests that sildenafil consumption can be detrimental in some cases [5,117,118,119,120,121]. For instance, Nivinson-Smith et al. tested the effects of sildenafil on visual function in mice heterozygous for the *rd1* mutation, which affects the PDE6 β-subunit [117]. The *rd1* mutation causes autosomal recessive retinitis pigmentosa, thereby carriers of the mutation do not display a disease phenotype. cGMP metabolism is altered in these individuals, rendering them more susceptible to retinal degeneration from external metabolic or oxidative stress. In their study, Nivinson-Smith et al. showed that sildenafil caused a significant dose-dependent decrease in photoreceptor ERG responses of wild type mice, which recovered within 48 h. However, decreased photoreceptor ERG responses of heterozygous *rd1* mice (*Pde6b^+/rd1^*) did not resolve until two weeks postadministration of the drug [117]. Behn et al. obtained very similar results using heterozygous PDE6 γ-subunit knockout mice (*Pde6g^+/tm1^*), another murine model of autosomal recessive retinitis pigmentosa [118]. Likewise, Pierce et al. administered a high dose of sildenafil citrate to dogs heterozygous for a functionally null mutation in PDE6 α-subunit (*Pde6a*) over a 4-month period. Despite the low number of animals used in their experiment, the results were statistically significant, showing that sildenafil-treated *Pde6a*^+/−^ dogs exhibited thinner outer nuclear layers and lower photoreceptor counts than untreated *Pde6a*^+/−^ dogs [119]. These data become especially relevant if we take into account that approximately 1 in 50 people are likely to be carriers of recessive traits leading to retinal degeneration. To date, no studies have been conducted in retinitis pigmentosa/cone-dystrophy patients or even in individuals who have normal vision but carry one allele for the disease. In a different paradigm, Eltony and Abdelhameed investigated the effect of chronic daily use of sildenafil on the histology of the retina and optic nerve of adult male rats and showed that sildenafil caused microglia activation, vacuolation and congested blood capillaries with apoptotic endothelial and pericytic cells, although partial recovery was observed after drug withdrawal [120]. Similar results were reported by Vatansever et al., who treated adult male rats with sildenafil for 4 weeks and observed dilatation and congestion in the choroidal vasculature, although no major changes were detected in retinal cytoarchitecture [121]. Therefore, in order to precisely exclude possible risks in these groups, it would be advisable to perform more research both at the preclinical and clinical levels.

Important regulatory agencies such as the U.S. Food and Drug Administration (FDA, www.fda.gov, accessed on 20 December 2020) and the European Medicines Agency (EMA, www.ema.europa.eu, accessed on 20 December 2020), and associations of eye physicians and surgeons such as the American Academy of Ophthalmology (www.aao.org, accessed on 20 December 2020) warn about the lack of controlled clinical data on the safety of sildenafil in patients with retinitis pigmentosa or with a family history of the disease. Thereby, it is essential that general practitioners supervise the treatment of a medical condition such as ED and guarantee a safe use of PDE5 inhibitors. The possibility of illegally purchasing online sildenafil and their analogues brings up relevant issues such as the risks linked to the irrational use of medicines. This is one of the factors that has prompted some countries to consider the reclassification of sildenafil from prescription-only medicine to a pharmacy medicine. Among the countries whose regulatory authorities have already taken that step are New Zealand in 2014 (Medicines and Medical Devices Safety Authority, Medsafe); the United Kingdom in 2017 (Medicines and Healthcare products Regulatory Agency, MHRA); Norway in 2019 (Norwegian Medicines Agency, NoMA); Ireland in 2020 (Health Products Regulatory Authority, HPRA). Although in these countries sildenafil can be sold without prescription, pharmacists receive specific training so they can provide proper guidance and request patients who answer some of the questions in the affirmative to contact their general practitioner for further assessment [122]. The aim of this practice is to lower the burden on general practitioners and, at the same time, to make the medication accessible while still keeping the risk of misuse and side effects low.

Finally, alternatives that minimize unwanted side effects should be pursued by scientists in general and by the pharmaceutical industry in particular. These may include, among others, the design, screening and development of drugs highly selective for PDE5 with no inhibitory effects on other PDE isoenzymes [123]; the investigation of new formulations that improve bioavailability [124]; the search of novel drug-delivery systems that allow a local vs. systemic application [29]; or the advancement in the field of pharmacogenomics, which would contribute to the implementation of a more precise and personalized medicine.

In conclusion, from the literature review we can affirm that visual side effects derived from the consumption of sildenafil are generally mild and transient, but the cessation of sildenafil therapy is advised if certain rare conditions such as central serous chorioretinopathy or NAION appear. Moreover, caution should be taken in patients with a family history of retinal dystrophy because available evidence in animal research supports the hypothesis that carriers of some recessive alleles are more sensitive to sildenafil toxicity.

## Figures and Tables

**Figure 1 biomedicines-09-00291-f001:**
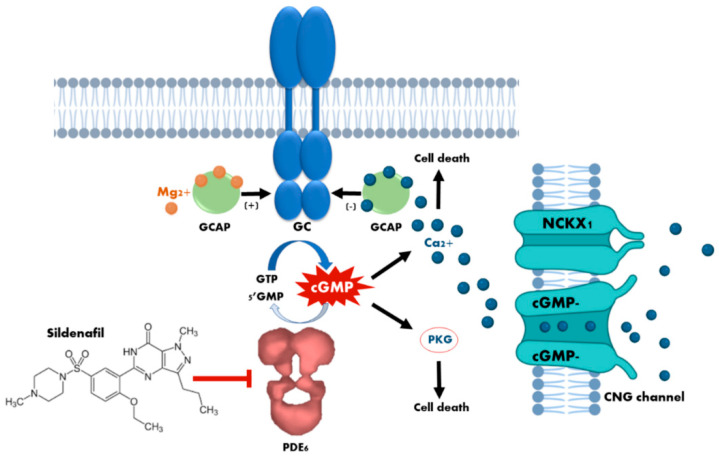
Cytotoxicity mechanisms of cyclic guanosine monophosphate (cGMP) in photoreceptors. The loss of phosphodiesterase (PDE6) function, or the prolonged activation of retinal guanylyl cyclase (GC) due to dominant mutations in the guanylyl cyclase-activating proteins (GCAPs), leads to an increase in the concentration of cytosolic cGMP which, in turn, causes continuous stimulation of the of protein kinase G (PKG) and an excessive influx of Ca^2+^ through the sustained opening of cyclic nucleotide-gated (CNG) channels. Both events have been demonstrated to cause photoreceptor cell death. The curved blue arrow represents cGMP synthesis by GC enzyme, whereas the curved white arrow represents cGMP hydrolisis by PDE6 enzyme. The blunt-end arrow represents the inhibition of PDE6 by sildenafil.

**Figure 2 biomedicines-09-00291-f002:**
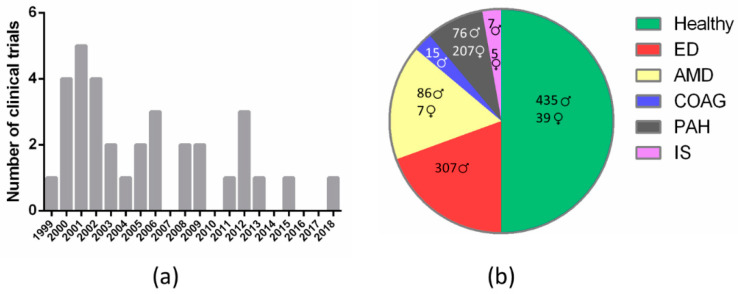
Classification of clinical trials that assess sildenafil effects on visual health according to: (**a**) year of publication (frequency histogram); (**b**) medical condition (pie chart; results are disaggregated by gender). ED, erectile dysfunction; AMD, age-related macular degeneration; COAG, chronic open-angle glaucoma; PAH, pulmonary arterial hypertension; IS, ischemic stroke.

**Table 1 biomedicines-09-00291-t001:** Common side effects of Phosphodiesterase type 5 (PDE5) inhibitors.

Side Effects	Sildenafil	Tadalafil	Vardenafil	Avanafil	Udenafil	Mirodenafil	Lodenafil
Headache	Yes	Yes	Yes	Yes	Yes	Yes	Yes
Flushing	Yes	Yes	Yes	Yes	Yes	Yes	Yes
Nasal congestion	Yes	Yes	Yes	Yes	Yes	Yes	Yes
Dyspepsia	Yes	Yes	Yes		Yes	Yes	Yes
Abnormal vision	Yes				Yes (only 200 mg)		Yes
Eye redness						Yes	
Sinusitis			Yes				
Flu syndrome							
Diarrhoea	Yes						
Myalgia		Yes					
Dizziness							Yes
Back pain		Yes	Yes				
Hyperemia					Yes		

**Table 2 biomedicines-09-00291-t002:** Summary of case reports/series on sildenafil-induced visual side effects published on the last decade (2010–2020).

Reference	Subject(s)	Dose	Risk Factors	Diagnosis
Felekis et al., 2011 [32] PMID 22034568	Man, age 51	Unknown, once a week for the last 6 months	Mild hypercholesterolemia Family history of NAION	Unilateral NAION (RE): decreased visual acuity, visual field loss, relative afferent pupillary defect, altered color perception, and optic disk edema
Moschos and Margetis, 2011 [33] PMID 21941503	Man, age 55	50 mg, 4–5 times a month for the last 8 months	None	Bilateral NAION: decreased visual acuity, visual field loss, relative afferent pupillary defect, and optic disk edema
Izadi et al., 2012 [43] PMID 22928790	Man, age 48	1500 mg, over a 4-hour period	None	Bilateral central visual field ring scotomas and reduced PERG amplitude
Tarantini et al., 2012 [44] PMID 22481954	Man, age 60	50 mg, 3 consecutive days	Noninsulin-dependent diabetes for the last 7 months, treated with metformin	Bilateral NAION: decreased visual acuity, visual field loss, optic disc edema, peripapillary nerve fiber layer hemorrhages, and serous macular detachment (only in RE)
Gaffuri et al., 2014 [45] PMID 24895393	Woman, 7-month-old infant	0.6 mg/kg/day in three doses	Preterm birth for maternal preeclampsia (34 weeks of gestation)	Bilateral NAION: sudden onset of visual loss with optic disc pallor, poor pupillary light reflex, arterial venous tortuous vessels, peripapillary retinal hemorrhages, and macular exudation
			Congenital heart defect	
Karli et al., 2014 [46] PMID 25378904	Man, age 42	Unknown	None	Unilateral atypical optic neuropathy (RE): vision loss, pain with ocular motility, optic disk edema, and optic nerve enhancement on MRI consistent with optic neuritis
Matheeussen et al., 2015 [47] PMID 26139313	Man, age 56	Overdose, 65 × 100 mg	None	Blurred vision and difficulties in distinguishing facial expressions. Subjective visual perception included a dark view with occasional light flashes
Coca et al., 2016 [48] PMID 27316292	Woman, age 39	3 × 20 mg a day, for the last 3 years	Bronchopulmonary dysplasia secondary to prematurity, PAH, kyphoscoliosis, pectus defect status postsurgery as an infant, severe obstructive and restrictive lung disease	Bilateral acute retrobulbar optic neuropathy attributable to PION
Jayadev et al., 2016 [49] PMID 27915325	Woman, premature infant (24.5 weeks of gestation)	0.8 mg/kg/day in 3 doses, starting on the 33rd week	Aggressive posterior retinopathy of prematurity and PAH	Bilateral retinal neovascularization in the eyes’ temporal quadrants, with hemorrhage in the LE
Sajjad and Weng, 2016 [50] PMID 27355186	Woman, age 32	3 × 20 mg a day, for the last 5 years	PAH and migraines	Bilateral asymmetrical outer macular atrophy: RPE mottling and atrophy in the RE, parafoveal RPE mottling and atrophy in a ring-like configuration, with decreased visual acuity in the LE
			Family history of PAH	
			Medications: topiramate, norethindrone, ambrisentan, tramadol, furosemide, pironolactone, and digoxin	
Li et al., 2018 [34] PMID 29487830	Woman, age 32	Overdose, 2000 mg	None	Color vision defects and blurred vision that resolved 38 days after drug uptake
Neufeld & Warner, 2018 [42] PMID 29215388	Man, age 66	Unknown, history of sildenafil use for 7 years, symptoms appear after using a “double dose”	Hypertension and hypercholesterolemia	Bilateral sequential NAION: visual field loss, relative afferent pupillary defect, and optic disk edema in the LE, with progressive visual acuity deterioration. One year later, the patient developed similar visual defects in the RE after using sildenafil 2 days in a row
Papageorgiou et al., 2018 [35] PMID 29374976	Man, age 56	Overdose, 40 × 100 mg	None	Retinal toxicity: decreased visual acuity, mild dilation of the retinal vessels, increased choroidal thickness, and persistent central ring scotomas on both eyes
Rickmann et al., 2018 [36] PMID 28776160	Man, age 53	50 mg, single dose	None	Acute unilateral loss of vision (RE)
Rosen et al., 2018 [37] PMID 30286227	Man, 57	100 mg, single dose	None	Photophobia and transient red-green deficiency. Colour perception improved 7 days after discontinuing sildenafil
Yanoga et al., 2018 [38] PMID 29489563	Man, age 31	Unknown (>50 mg/mL), single dose	None	Multicolor photopsias, erythropsia, subjective sense of decreased contrast, increased choroidal thickness, and outer retina disruptions
Brader et al., 2019 [39] PMID 30629106	Man, age not specified (mid 50s)	750 mg, single dose	None	Photophobia, nyctalopia, bilateral central ring-shaped scotomas, and outer retina disruptions
Mohammadpour et al., 2019 [40] PMID 31372081	Man, age 35	4 × 100 mg in a three-day period	None	Unilateral central serous chorioretinopathy (LE): decreased vision, metamorphopsia, altered colour perception, loss of foveal reflex, serous retinal detachment in the foveal region, and increased foveal thickness
Karaarslan, 2020 [41] PMID 32117027	17 men, age 38–57	100 mg, single dose	None of the 17 patients had a history of ocular pathology (including glaucoma) or any diagnosed systemic disease	52.9% exhibited some degree of clinical photophobia, 76.5% had altered colour vision, 17.6% had a deficiency in stereopsis, 35.3% had a transient contrast sensitivity impairment, and 47.1% had abnormally dilated pupils although no relative afferent defects were found

NAION: nonarteritic anterior ischemic optic neuropathy; RE: right eye; LE: left eye; PERG: pattern electroretinogram; PION: posterior ischemic optic neuropathy; RPE: retinal pigment epithelium; PAH: pulmonary arterial hypertension.

**Table 3 biomedicines-09-00291-t003:** Summary of studies designed to assess the effects of sildenafil on vision.

ID Number (Publication Year)	Reference	Participants	Design + Dose (PC/OL)	Assessments	Results
CN-00675062 (2009)	Laties et al., 1999 [59]	Phase I trial:	PC, acute study	Color discrimination	Statistically significant increase in FM 100-Hue total error scores, 1–2 h after sildenafil consumption (100 or 200 mg). Fully reversible effects that coincided with peak plasma sildenafil concentrations
		16 healthy men	50–200 mg	(FM 100-Hue)	
		(age not available)			
		Phase II-III trial:	OL, 12–40 weeks	Color discrimination	Nonsignificant clinical changes in FM 100-Hue test at 12 or 52 weeks compared with baseline measurements
		47 men with ED	25–200 mg	(FM 100-Hue)	
		(age not available)			
CN-00679125 (2009)	Hoffman et al., 2000 [74]	8 healthy volunteers	PC, acute study, single oral dose (dose not available)	HVF test	Quadrantanopic field defects in the Humphrey visual field test were reported in only one subject. Nondetectable changes in blue-on-yellow or white-on-white Humphrey visual field test after sildenafil consumption
		(age and sex not available)			
CN-00329981 (2000)	McCulley et al., 2000 [55]	8 healthy volunteers (20–38 years)	PC, acute study	HVF test	One of five subjects in the sildenafil group performed poorly on HVF testing, with blue-on-yellow affected more than white-on-white
	PMID 11130743		200 mg		
CN-00297288 (2000)	Yajima et al., 2000 [75]	48 healthy men (age not available)	PC, acute study	IOP and pupil diameter	Nonsignificant clinical changes were observed in IOP or pupil diameter after administration of sildenafil
	PMID 10844068		10–150 mg		
CN-00674478 (2009)	Zrenner et al., 2000 [58]	Phase II trial:	OL, 40 weeks	Visual acuity, color discrimination, contrast sensitivity, photostress test and slit-lamp examination	Nonsignificant clinical changes after 2 years of sildenafil consumption in any of the visual tests or eye structure examinations. No discontinuations due to visual adverse events
		48 men with ED	25–100 mg		
		(age not available)			
		Phase III trial:	OL, 2 years		
		31 men with ED	25–100 mg		
		(age not available)			
CN-00379913 (2003)	Dündar et al., 2001a [76]	40 healthy men (21–32 years sildenafil group; 20–30 years placebo group)	PC, acute study	Resting heart rate, blood pressure, ECG, visual acuity and color vision	No ocular effects were described during the treatment period with sildenafil. Significantly increased heart rate after sildenafil administration compared with baseline. Nonsignificant decrease in blood pressure. Common side effects such as flushing, headache, dyspepsia, unintentional incomplete sexual arousal and palpitation were increased in the sildenafil group
	PMID 11989569		50 mg as a single oral dose		
	Dündar et al., 2001b [57]	14 healthy men (20–38 years)	Acute study	Visual acuity, IOP, vision, anterior segment, fundus appearance, resting heart rate,	Statistically significant increase in heart rate after sildenafil administration compared with baseline. No other changes on visual acuity, color vision, lOP, systolic blood pressure or diastolic blood pressure were observed
	PMID 11767027		50 mg as a single oral dose	blood pressure and blood flow (color Doppler imaging)	
CN-00348336 (2001)	Grunwald et al., 2001a [77]	15 healthy men (39 ± 8 years)	PC, acute study	Blood pressure, IOP and perfusion pressure	No significant changes in mean blood pressure, IOP, perfusion pressure, choroidal or optic nerve circulatory parameters were observed after sildenafil treatment
	PMID 11384572		100 mg on 2 separate days		
CN-00375859 (2003)	Grunwald et al., 2001b [78]	15 men with bilateral COAG (49–77 years)	PC, acute study	IOP, brachial artery blood pressure and heart rate	No statistically or clinically significant changes in IOP, mean systemic blood pressure or heart rate was detected after sildenafil treatment
	PMID 11730651		100 mg on 2 separate days		
CN-00676836 (2009)	Luu et al., 2001 [79]	18 healthy volunteers: 12 men and 6 women (18–55 years)	PC, acute study	Color discrimination (Lanthony desaturated panel D-15) and ERG	Statistically significant increase in visual disturbances: changed light sensitivity, blurred vision, after-images and color axis. Small changes in cone function (modestly depressed in both, macula and periphery). ERG parameters were within the normal limits
	PMID 11530053		200 mg as a single oral dose		
CN-00379944 (2003)	Birch et al., 2002 [56]	9 men with early-AMD (59–85 years)	PC, acute study	Visual acuity, Amsler grid, color discrimination, HVF test and photostress test	No statistically or clinically relevant acute changes in any visual function test compared with no drugs. No visual adverse effects
	PMID 11992864		100 mg		
CN-00674741 (2009)	Friedman et al., 2002 [80]	6 healthy men (age not available)	PC, acute study	Optical properties of the eye (measured by Shack-Hartman wavefront sensing) and contrast sensitivity	Significant shift in defocus consistent with an anterior movement in retinal location, with attendant increase in contrast sensitivity from baseline
			50 mg as a single oral dose		
CN-00380294 (2003)	Grunwald et al., 2002 [81]	15 healthy men (31–47 years)	PC, acute study	Monochromatic fundus photography, brachial artery blood pressure, IOP and diameters of two major temporal veins and one artery	Statistically nonsignificant changes in average venous diameter were observed for the superior and the inferior retinal temporal veins, or the retinal temporal artery were reported after sildenafil treatment. No significant differences in the percentage change from baseline in venous or arterial diameter at 1 or 5 h after sildenafil consumption
CN-00793125 (2011)	PMID 12036673		100 mg on 2 separate days		
CN-00413065 (2003)	McCulley et al., 2002 [82]	13 healthy volunteers: 4 men and 9 women (23–49 years)	PC, acute study	Choroidal thickness, color discrimination and contrast sensitivity	Slight changes (statistically nonsignificant) in color discrimination, error scores increased after sildenafil consumption. Nonsignificant changes in choroidal thickness and contrast sensitivity relative to baseline in either group
	PMID 12566892		200 mg		
CN-00717900 (2009)	Mollon et al., 2003 [83]	16 young healthy men (age not available)	PC, acute study	Visual persistence	Statistically significant increase in interstimulus interval value after ingestion of 100 mg of sildenafil, compared with no drugs. The effects were fully reversible
			25–200 mg		
CN-00458473 (2004)	Polak et al., 2003 [84]	12 healthy men (36–59 years)	PC, acute study	Retinal vessel diameters, retinal blood velocity, response of retinal vessel diameters to flicker stimulation, blood pressure and IOP	Nonsignificant effects on mean arterial pressure, pulse rate, IOP, retinal blood velocity, retinal arterial diameter, or flicker-induced vasodilation. Significant increase in retinal venous diameters and retinal blood flow
	PMID 14578411		100 mg as a single oral dose		
CN-00469626 (2004)	Jägle et al., 2004 [62]	20 healthy men (20–40 years)	PC, acute study	ERG, contrast sensitivity and color vision	Statistically significant changes in contrast sensitivity during transient visual effect (tritanopia) and in ERG. Nonsignificant differences in color discrimination. No visual adverse effects were reported. Acute effects were fully reversible within 24 h
	PMID 15126148		100 mg as a single oral dose		
CN-00511642 (2005)	Koksal et al., 2005 [85]	30 men with ED (23–74 years sildenafil group; 21–56 years placebo group)	PC, acute study	IOP, systolic and diastolic blood pressure and ocular blood flow	Significant increase in blood flow in the ophthalmic artery and the short posterior ciliary artery
	PMID 15948790		100 mg as a single dose.		
CN-00523673 (2006)	Metelitsina et al., 2005 [86]	15 men with AMD (68–82 years)	PC, acute study	Relative choroidal blood velocity, volume and flow, BCVA, contrast sensitivity, mean arterial blood pressure, heart rate, IOP and ocular perfusion pressure	Significant decreases in mean arterial blood pressure and perfusion pressure were observed 30 min after sildenafil administration but no statistically significant changes in foveolar choroidal circulation of AMD patients were found. Nonsignificant changes in BCVA, contrast sensitivity, IOP or heart rate were described
	PMID 16080909		100 mg on 2 separate days		
	Dündar et al., 2006a [54]	15 men with ED (33–60 years)	OL, 3 months	BCVA, IOP, color vision, slit-lamp examination, funduscopy and blood flow (color Doppler imaging)	No ocular effects of sildenafil were considered statistically significant compared with the baseline. No visual abnormalities were reported after sildenafil administration
	PMID 16292333		50 mg twice a week		
	Dündar et al., 2006b [87]	14 men with ED (35–60 years)	OL, 3 months	BCVA, color vision, IOP, funduscopy and HVF test	No significant changes in BCVA, color vision, and IOP were observed after sildenafil treatment compared with baseline. There was no change on blue-on-yellow and white-on-white Humphrey perimetry tests
	PMID 16052253		50 mg twice a week		
CN-00563219 (2007)	Metelitsina et al., 2006 [88]	14 men with AMD (68–82 years)	PC, acute study	Diameter of the major retinal veins	Sildenafil citrate produces a statistically significant vasodilatation of major retinal veins
	PMID 16530757		100 mg on 2 separate days		
CN-01514954 (2018)	Bayer 2007 [89]	63 healthy men	PC, acute study Sildenafil 200 mg/day for 2 days	Color discrimination and ERG	Not available
NCT00461565		(18–55 years)	PC, 8 weeks Vardenafil 20 mg twice a week		
CN-00699940 (2009)	Foresta et al., 2008 [63]	30 healthy men (24–33 years)	PC, acute study	Blood flow velocity in the ophthalmic artery	Effect in a time-dependent manner. Statistically significant increased blood flow velocity from baseline in the ophthalmic artery, 60 min after drug uptake. No changes were reported from 4 to 36 h after drug administration.
	PMID 17585311		100 mg sildenafil, 20 mg tadalafil		
CN-00754551 (2009)	Ibrahim et al., 2008 [90]	40 men with early AMD (55–86 years)	PC, acute study	Visual acuity, Amsler grid, and color discrimination	Statistically nonsignificant changes in visual acuity or color discrimination, compared with no drugs
			100 mg		
CN-00687939 (2009)	Cordell et al., 2009 [91]	244 healthy men or with mild ED men (30–65 years)	PC, 6 months	ERG, visual acuity, color discrimination, HVF test, slit-lamp examination, funduscopy and IOP	Nonsignificant clinical changes in ophthalmologic examinations and visual tests between the sildenafil group and the placebo group after 6 months of treatment
CN-02013932 (2020)	PMID 19365010		Sildenafil 50 mg/day		
NCT00333281			Tadalafil 5 mg/day		
CN-01598223 (2018)	Silver et al., 2009 [92]	12 patients: 7 men and 5 women with IS (18–80 years)	OL, 2 weeks	Stroke worsening, new stroke, myocardial	No ocular effects were described during the treatment period with sildenafil
NCT00452582	PMID 19717023		25 mg on a daily basis	Infarction, vision, and hearing loss	
CN-00799471 (2012)	Gerometta et al., 2011 [65]	9 healthy volunteers: 6 men and 3 women (18–74 years)	PC, acute study 100 mg	IOP and blood pressure	Statistically significant transient IOP increase that resolved within 2 h from sildenafil administration. Both systolic and diastolic blood pressures were significantly reduced by sildenafil and this effect persisted throughout 2 h
	PMID 21651908				
NCT01642407	Pfizer 2012 [93]	6 men with PAH (1–17 years)	OL, 4 weeks up to a maximum of 119.6 weeks.	External examination of the eye, slit-lamp examination, funduscopy, visual acuity and color vision	No ocular effects of sildenafil were reported during the study
			Body weight-dependent dose (>20 kg: 20 mg thrice a day; ≤20 kg: 10 mg thrice a day)		
CN-00833380 (2012)	Wirostko et al., 2012 [94]	277 patients with PAH: 70 men and 207 women (age not available)	PC, 12 weeks	External inspection of the eye, slit-lamp examination, funduscopy, IOP, BCVA, contrast sensitivity, color vision and HVF test	Nonsignificant clinical changes in the ophthalmic examinations and visual tests were reported, but deterioration in visual acuity from baseline to week 12 ranged from 10% in the placebo group to 3% in the 20 mg sildenafil group. A modest, dose-related incidence of chromatopsia, cyanopsia, photophobia, and visual disturbance was reported in the 80 mg sildenafil group
NCT00644605 (PC)	PMID 22354598		20, 40 and 80 mg, three times daily		
NCT00159887 (OL extension)					
		259 patients with PAH (222 of those completed 1 year of treatment)	OL, up to 3 years		At week 24, nonsignificant clinical changes were observed in comparison with week 12 results. Low incidence of ocular and transient adverse events (<0.5–10%), which decreased as the study progressed
			40 mg three times daily during 6 weeks + 80 mg up to 3 months.		
			After 3 months, sildenafil was titrated according to clinical need (max. 80 mg and min. 20 mg three times daily) for 3 years		
NCT01830790	Duke University 2013 [95]	10 AMD patients: 6 men and 4 women	Acute study	Visual acuity, choroidal thickness, central macular thickness and macular volume	Not available
		(≥65 years)	100 mg		
NCT02364882	Strategic Science & Technologies, LLC 2015 [96]	21 healthy postmenopausal women (35–65 years)	OL, 50, 100 and 200 mg sildenafil 50% external (i.e., labia minora and clitoral area)/50% intravaginal	Safety and pharmacokinetic profile of topical sildenafil administration	Adverse ocular effects reported: chromatopsia, increased sensitivity to light and blurred vision
CN-01614537 (2018)	Coleman et al., 2018 [97]	5 patients: 2 men and 3 women with AMD	OL, 2 years	BCVA, contrast sensitivity, OCT, angiography and funduscopy	Slight beneficial effects in vision measured by BCVA, although nonsignificant clinical changes after 2 years of sildenafil consumption
	PMID 29694963	(29–73 years)	body weight-dependent dose (≤150 lbs: 20 mg twice a day; >150 lbs: up to 40 mg twice a day)		

AMD, age-related macular degeneration; BCVA, best-corrected visual acuity; ECG, electrocardiogram; ED, erectile dysfunction; ERG, electroretinogram; IOP, intraocular pressure; OCT, optical coherence tomography; PC, placebo-controlled trial; OL, open-label trial; FM: Farnsworth-Munsell; HVF: Humphrey visual field; PAH: pulmonary arterial hypertension; IS: ischemic stroke; COAG: chronic open-angle glaucoma.

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
