# Peer review of "Visual Side Effects Linked to Sildenafil Consumption: An Update"

_biomedicines, 2021, doi:10.3390/biomedicines9030291_

Round 1
Reviewer 1 Report
1) Need to add detailed pharmacological offers.
2) Need to add Cyp450 metabolism and drug metabolism details.
3) Need to improve formatting and provide informative details in tabular format.
Author Response
1) Need to add detailed pharmacological offers.
We acknowledge the reviewer’s suggestion and have added new information regarding sildenafil analogues and different formulations/drug delivery systems. This can be found in section 1.2. Phosphodiesterases and inhibitors:
<<In addition to the three drugs mentioned above, the second-generation inhibitor avanafil (Stendra®) became internationally available in 2013. Avanafil exhibited 100-times lower specificity for PDE6 than for PDE5, presumably reducing the potential side effects derived from the non-selective inhibition of PDE6 by sildenafil and vardenafil (reviewed in Zucchi et al 2019). Other second-generation (udenafil and mirodenafil) or third-generation (lodenafil, SLX-2101, JNJ-10280205, and JNJ-10287069) PDE5 inhibitors have been either approved and introduced into the market in some parts of the world or are at the final stages of their clinical development. Udenafil (Zydena) is only available in some Asian countries and Russia, mirodenafil (Mvix) is commercialized in various Asian countries and lodenafil (Helleva) is sold in Brazil (Hatzimouratidis et al, 2016). All of them have been trialled in tablet formulation at different doses, whose broadest range spans from 25mg to 200mg (Grice et al, 2020). Several studies indicate that, in general terms, PDE5 inhibitors are well tolerated and their side effects are few, mild and very similar among the different compounds studied, except for tadalafil, which caused a higher incidence of myalgia (Table 1). Many of the side effects are due to the vasoactivity of these compounds, given the expression of PDE5 in vascular smooth muscles. The most common reported dose-dependent adverse events include headache, flushing, nasal congestion, facial and ocular hyperemia, myalgia, back pain and dyspepsia (Ferguson et al, Corona et al 2016; Anderson et al, 2018)>>.
<<The occurrence of side effects increases with both serum levels and exposure time to the PDE5 inhibitor (Taylor et al., 2009). To overcome these issues, novel drug formulations that improve the safety and efficacy profile of the drug are under development. Despite the side effects, oral administration of PDE5 inhibitors (tablets, oral solution or orodispersible tablets) is nowadays considered the first-line therapy for ED. A second-line treatment consists of invasive procedures such as intracavernosal injections with vasogenic drugs like alprostadil (synthetic prostaglandin E1), papaverine or phentolamine, as well as intraurethral alprostadil suppositories and vacuum erection devices. These show a more favourable systemic side effect profile compared to oral pharmacotherapy (Belew et al., 2015) and, despite being invasive, they are widely used. To avoid invasive techniques and, at the same time, minimize systemic side effects, topical formulations (alprostadil and sildenafil topical cream) constitute a promising alternative, as they can be applied locally, are safe and easy to use (Patel et al., 2016; Grice et al., 2020; Kim et al., 2021). Also, solid lipid nanoparticles in hydrogel films for the transdermal local delivery of avanafil have been assayed in vitro and ex vivo with success (Kurakula 2016). Moreover, emerging medications and procedures are currently under investigation for the treatment of ED in both preclinical and clinical settings, including non-PDE5 inhibitors oral drugs such as melanocortin receptor antagonists (subcutaneous melanocortin analogue (PT-141)), rho-kinase inhibitors (SAR407899), and soluble GC activator (BAY60-4552 and BAY 60-2770) (Patel et al., 2016; Grice et al., 2020; Kim et al., 2021). Also under consideration are regeneration therapy involving stem cell injection; gene therapy where the genetic material can be easily injected into the penis; low-intensity extracorporeal shock wave therapy; low-intensity pulse ultrasound; and platelet-rich plasma injections (Patel et al., 2016; Grice et al., 2020; Kim et al., 2021). Finally, the use of nanotechnology for drug delivery is being studied in murine models for all delivery methods (oral, topical, and intracavernosal), as a way to either enhance bioavailability or to improve and promote the local effects of medications (Wang et al., 2017)>>.
2) Need to add Cyp450 metabolism and drug metabolism details.
We acknowledge the reviewer’s suggestion and have added a new paragraph in section 1.3. Side effects of sildenafil:
<<Besides, sildenafil pharmacokinetics can be modified by the concomitant use of other drugs such as inhibitors of the cytochrome P450 (CYP) 3A4 (e.g. macrolide antibiotics, calcium channel blockers, etc.) (Dresser et al., 2000), which is the main enzyme responsible for its hepatic metabolization. Inhibition of CYP3A4 would elevate the plasma concentration of sildenafil, thereby also increasing the likelihood of unwanted side effects. These key drug-metabolizing enzymes often display genetic polymorphisms that contribute to the individual variability in drug safety and efficacy among patients and may represent a risk of drug-drug interactions (Tang et al., 2020)>>.
Also, we have added a brief comment in the Discussion:
<<Among the different factors that could influence individual sensitivity to sildenafil are gene polymorphisms of CYP3A4 and CYP2C9, the two major sildenafil-metabolizing hepatic enzymes (Tang et al, 2020)>>.
3) Need to improve formatting and provide informative details in tabular format.
Indeed, the tabular format is something that we had already considered in our first version of the manuscript but, due to a mistake when transferring the manuscript from a different MDPI journal to Biomedicines, the table was missing in the submitted version. We regret this misunderstanding and have incorporated the table in the revised version. Please see Table 2.
Reviewer 2 Report
The topic is new and futuristic and no similar thematic review articles have been published till date. Does correlate and give significant information on sildenafil consumption related adverse effects especially on eye sight. However, extensive English language revisions are required. The review certainly does add scrutiny in the respective field of research. Further, few minor revisions are required in order for the review to be accepted for publication. 1. Abstract does not highlight why studies on sildenafil consumption related adverse effects are important. Even does not discuss the future research or outlook of this type of study. 2. A table is recommended for grouping the methodology or experimental design- separately for side effects and Ophthalmologic examinations 3. The discussions are not example based does not connect properly with just directional explanation. Authors are advised to add certain relevant examples so that readers get a detailed information. 4. A table listing the novel drug delivery systems being prepared to overcome PDE-5 inhibitors side/adverse effects are reported in literature in recent times and these need to be added and is strongly recommended for the review. 5. The review lacks upbringing the importance of irrational use or medication under physician direction concept to minimize and their importance in highlighting to build and safe use of PDE-5. New references suggested below should be included in the discussion. 6. A new section or table on listing all the PDE-5 inhibitors and their possible side effects. 7. A detailed explanation on how this correlation is important for different stakeholders such as healthcare professionals, patients, scientist and pharmaceutical industries needs to be commented in the conclusion section. 8. The review lacks extraction of valuable figures or data from the references cited. Authors are strongly suggested for the needful. 9. Also below are the few latest papers /case studies on PDE-5 delivery systems developed and evaluated to minimize side effects, authors are strongly advised to cite and mention in the respective discussion sections. Authors are strongly suggested for these recommendations to be considered for further evaluation. Clinical & experimental ophthalmology 37, no. 5 (2009): 514-523. Journal of Liposome Research 26, no. 4 (2016): 288-296. Drug safety 32, no. 1 (2009): 1-18. Pharmaceutics 12, no. 11 (2020): 1124. The journal of sexual medicine 8, no. 10 (2011): 2894-2903. Ophthalmologica 227, no. 2 (2012): 85-89. Nippon Ganka Gakkai Zasshi 101, no. 7 (1997): 551-557.Author Response
The topic is new and futuristic and no similar thematic review articles have been published till date. Does correlate and give significant information on sildenafil consumption related adverse effects especially on eye sight. However, extensive English language revisions are required. The review certainly does add scrutiny in the respective field of research. Further, few minor revisions are required in order for the review to be accepted for publication.
We acknowledge the encouraging comments. We have taken into consideration the reviewer’s suggestions and now the revised version of the manuscript is much-improved thanks to them.
- Abstract does not highlight why studies on sildenafil consumption related adverse effects are important. Even does not discuss the future research or outlook of this type of study.
According to the reviewer’s suggestion, the abstract has been modified to highlight these important issues.
- A table is recommended for grouping the methodology or experimental design- separately for side effects and Ophthalmologic examinations.
In our first version of the manuscript, we presented the data in a tabular format but, due to a mistake when transferring the manuscript from a different MDPI journal to Biomedicines, the table was missing in the submitted version. We regret this misunderstanding and have incorporated the original table in the revised version (please see Table 2). We think that presenting the data in this format will probably resolve the reviewer’s concerns.
- The discussions are not example based does not connect properly with just directional explanation. Authors are advised to add certain relevant examples so that readers get a detailed information.
According to the reviewer’s suggestion, we have given detailed examples of other researchers’ related work in the field and we hope this makes more clear our point of view regarding the lack of controlled clinical data in some populations. A whole new paragraph can be found in the Discussion:
<<Besides, evidence from preclinical studies carried out in animal models of human retinal diseases suggest that sildenafil consumption can be detrimental in some cases [5,86–89]. For instance, Nivinson-Smith et al. tested the effects of sildenafil on visual function in mice heterozygous for the rd1 mutation, which affects PDE6 β-subunit (Nivinson-Smith et al, 2014). The rd1 mutation causes autosomal recessive retinitis pigmentosa, thereby carriers of the mutation do not display a disease phenotype. cGMP metabolism is altered in these individuals though, rendering them more susceptible to retinal degeneration from external metabolic or oxidative stress. In their study, Nivinson-Smith et al. showed that sildenafil caused a significant dose-dependent decrease in photoreceptor ERG responses of wild type mice, which recovered within 48 hours. However, decreased photoreceptor ERG responses of heterozygous rd1 mice (Pde6b+/rd1) did not resolve until two weeks post administration of the drug (Nivinson-Smith et al, 2014). Behn and Potter obtained very similar results using heterozygous PDE6 γ-subunit knockout mice (Pde6g+/tm1), another murine model of autosomal recessive retinitis pigmentosa (Behn and Potter, 2000). Likewise, Pierce et al. administered a high dose of sildenafil citrate to dogs heterozygous for a functionally null mutation in PDE6 α-subunit over a 4-month period. Despite the low number of animals used in their experiment, the results were statistically significant, showing that sildenafil-treated Pde6a+/− dogs exhibited thinner outer nuclear layer and lower photoreceptor counts than untreated Pde6a+/− dogs (Pierce et al, 2019). These data become especially relevant if we take into account that approximately 1 in 50 people are likely to be carriers of recessive traits leading to retinal degeneration. To date, no studies have been conducted in retinitis pigmentosa/cone-dystrophy patients or even in individuals who have normal vision but carry one allele for the disease. On a different paradigm, Eltony and Abdelhameed investigated the effect of chronic daily use of sildenafil on the histology of the retina and optic nerve of adult male rats and showed that sildenafil caused microglia activation, vacuolation and congested blood capillaries with apoptotic endothelial and pericytic cells, although partial recovery was observed after drug withdrawal (Eltony and Abdelhameed, 2017). Similar results were reported by Vatansever et al., who treated adult male rats with sildenafil for 4 weeks and observed dilatation and congestion in the choroidal vasculature, although no major changes were detected in retinal cytoarchitecture (Vatansever et al. 2003). Therefore, in order to precisely exclude possible risks in these groups, it would be advisable to perform more research both at the preclinical and clinical level>>.
- A table listing the novel drug delivery systems being prepared to overcome PDE-5 inhibitors side/adverse effects are reported in the literature in recent times and these need to be added and is strongly recommended for the review.
According to the reviewer’s suggestion, we have incorporated a new paragraph in section 1.2. Phosphodiesterases and inhibitors, regarding novel drug delivery systems. Consequently, new references have been added to the list.
<<The occurrence of side effects increases with both serum levels and exposure time to the PDE5 inhibitor (Taylor et al., 2009). To overcome these issues, novel drug formulations that improve the safety and efficacy profile of the drug are under development. Despite the side effects, oral administration of PDE5 inhibitors (tablets, oral solution or orodispersible tablets) is nowadays considered the first-line therapy for ED. A second-line treatment consists of invasive procedures such as intracavernosal injections with vasogenic drugs like alprostadil (synthetic prostaglandin E1), papaverine or phentolamine, as well as intraurethral alprostadil suppositories and vacuum erection devices. These show a more favourable systemic side effect profile compared to oral pharmacotherapy (Belew et al., 2015) and, despite being invasive, they are widely used. To avoid invasive techniques and, at the same time, minimize systemic side effects, topical formulations (alprostadil and sildenafil topical cream) constitute a promising alternative, as they can be applied locally, are safe and easy to use (Patel et al., 2016; Grice et al., 2020; Kim et al., 2021). Also, solid lipid nanoparticles in hydrogel films for the transdermal local delivery of avanafil have been assayed in vitro and ex vivo with success (Kurakula 2016). In addition, emerging medications and procedures are currently under investigation for the treatment of ED in both preclinical and clinical settings, including non-PDE5 inhibitors oral drugs such as melanocortin receptor antagonists (subcutaneous melanocortin analogue (PT-141)), rho-kinase inhibitors (SAR407899), and soluble GC activator (BAY60-4552 and BAY 60-2770) (Patel et al., 2016; Grice et al., 2020; Kim et al., 2021). Also under consideration are regeneration therapy involving stem cell injection; gene therapy where the genetic material can be easily injected into the penis; low-intensity extracorporeal shock wave therapy; low-intensity pulse ultrasound; and platelet-rich plasma injections (Patel et al., 2016; Grice et al., 2020; Kim et al., 2021). Finally, the use of nanotechnology for drug delivery is being studied in murine models for all delivery methods (oral, topical, and intracavernosal), as a way to either enhance bioavailability or to improve and promote the local effects of medications (Wang et al., 2017)>>.
- The review lacks upbringing the importance of irrational use of medication under physician direction concept to minimize and their importance in highlighting to build and safe use of PDE-5. New references suggested below should be included in the discussion and 7. A detailed explanation of how this correlation is important for different stakeholders such as healthcare professionals, patients, scientist and pharmaceutical industries needs to be commented on in the conclusion section.
We agree with the reviewer that our previous version lacked upbringing the importance of irrational use or medication. It was our intention from the beginning to transmit this idea but we failed to do so. In the revised version of the manuscript we have added new paragraphs in the Discussion where we address the reviewer’s concerns regarding points 5) and 7):
<<Important regulatory agencies such as the U.S. Food and Drug Administration (FDA, www.fda.gov) and the European Medicines Agency (EMA, www.ema.europa.eu), and associations of eye physicians and surgeons like the American Academy of Ophthalmology (www.aao.org) warn about the lack of controlled clinical data on the safety of sildenafil in patients with retinitis pigmentosa or with a family history of the disease. Thereby it is essential that general practitioners supervise the treatment of a medical condition like ED and guarantee the safe use of PDE5 inhibitors. The possibility of illegally purchasing online sildenafil and their analogues brings up relevant issues such as the risks linked to the irrational use of medicines. This is one of the reasons that have prompted some countries to consider the reclassification of sildenafil from a prescription-only medicine to pharmacy medicine. Among the countries whose regulatory authorities have already taken that step are New Zealand in 2014 (Medicines and Medical Devices Safety Authority, Medsafe); the United Kingdom in 2017 (Medicines and Healthcare products Regulatory Agency, MHRA); Norway in 2019 (Norwegian Medicines Agency, NoMA); and Ireland in 2020 (Health Products Regulatory Authority, HPRA). Although in these countries sildenafil can be sold without a prescription, pharmacists receive specific training so they can provide proper guidance and request patients who answer some of the questions in the affirmative to contact their general practitioner for further assessment (Braund 2018). The aim of this practice is to lower the burden on general practitioners and, at the same time, to make the medication accessible while still keeping the risk of misuse and side effects low.
Finally, alternatives that minimize unwanted side effects should be pursued by scientists in general and by the pharmaceutical industry in particular. These may include, among others, the design, screening and development of drugs highly selective for PDE5 with no inhibitory effects on other PDE isoenzymes (Kayik 2017); the investigation of new formulations that improve bioavailability (Hosny 2020); the search of novel drug-delivery systems that allow a local vs. systemic application (Kurakula 2016); or the advancement in the field of pharmacogenomics, which would contribute to implementing a more precise and personalized medicine.
In conclusion, from the literature review, we can affirm that visual side effects derived from the consumption of sildenafil are generally mild and transient, but it is advised cessation of sildenafil therapy if certain rare conditions such as central serous chorioretinopathy or NAION appear. Moreover, caution should be taken in patients with a family history of retinal dystrophy, because available evidence in animal research supports the hypothesis that carriers of some recessive alleles are more sensitive to sildenafil toxicity>>.
- A new section or table on listing all the PDE-5 inhibitors and their possible side effects.
According to the reviewer’s suggestion, we have incorporated a new paragraph and a table (Table 1) in section 1.2. Phosphodiesterases and inhibitors:
<<In addition to the three drugs mentioned above, the second-generation inhibitor avanafil (Stendra®) became internationally available in 2013. Avanafil exhibited 100-times lower specificity for PDE6 than for PDE5, presumably reducing the potential side effects derived from the non-selective inhibition of PDE6 by sildenafil and vardenafil (reviewed in Zucchi et al 2019). Other second-generation (udenafil and mirodenafil) or third-generation (lodenafil, SLX-2101, JNJ-10280205, and JNJ-10287069) PDE5 inhibitors have been either approved and introduced into the market in some parts of the world or are at the final stages of their clinical development. Udenafil (Zydena) it is only available in some Asian countries and Russia, mirodenafil (Mvix) is commercialized in various Asian countries and lodenafil (Helleva) is sold in Brazil (Hatzimouratidis et al, 2016). All of them have been trialled in tablet formulation at different doses, whose broadest range spans from 25mg to 200mg (Grice et al, 2020). Several studies indicate that, in general terms, PDE5 inhibitors are well tolerated and their side effects are few, mild and very similar among the different compounds studied, except for tadalafil, which caused a higher incidence of myalgia (Table 1). Many of the side effects are due to the vasoactivity of these compounds, given the expression of PDE5 in vascular smooth muscles. The most common reported dose-dependent adverse events include headache, flushing, nasal congestion, facial and ocular hyperemia, myalgia, back pain and dyspepsia (Ferguson et al, Corona et al 2016; Anderson et al, 2018)>>.
Table 1. Common side effects of PDE5 inhibitors
Side effects |
Sildenafil |
Tadalafil |
Vardenafil |
Avanafil |
Udenafil |
Mirodenafil |
Lodenafil |
Headache |
✔ |
✔ |
✔ |
✔ |
✔ |
✔ |
✔ |
Flushing |
✔ |
✔ |
✔ |
✔ |
✔ |
✔ |
✔ |
Nasal congestion |
✔ |
✔ |
✔ |
✔ |
✔ |
✔ |
✔ |
Dyspepsia |
✔ |
✔ |
✔ |
|
✔ |
✔ |
✔ |
Abnormal vision |
✔ |
|
|
|
✔(only 200 mg) |
|
✔ |
Eye redness |
|
|
|
|
|
✔ |
|
Sinusitis |
|
|
✔ |
|
|
|
|
Flu syndrome |
|
|
|
|
|
|
|
Diarrhoea |
✔ |
|
|
|
|
|
|
Myalgia |
|
✔ |
|
|
|
|
|
Dizziness |
|
|
|
|
|
|
✔ |
Back pain |
|
✔ |
✔ |
|
|
|
|
Hyperemia |
|
|
|
|
✔ |
|
|
- The review lacks extraction of valuable figures or data from the references cited. Authors are strongly suggested for the needful.
In the new version of the manuscript, we have incorporated tables 1 and 2, which gather the most common side effects of commercialized PDE5 inhibitors (Table 1) and all the relevant findings of the clinical trials (Table 2). Furthermore, we have now added more detailed explanations of some issues requested by the referees. Additionally, we provide two figures that we think are valuable: Figure 1 helps understand the critical role cGMP plays in the visual transduction pathway and the negative consequences of its elevation, and Figure 2 gives a global vision of the number of clinical trials conducted to evaluate sildenafil side effects related to vision, including data related with gender and previous disease conditions.
- Also below are the few latest papers /case studies on PDE-5 delivery systems developed and evaluated to minimize side effects, authors are strongly advised to cite and mention in the respective discussion sections. Authors are strongly suggested for these recommendations to be considered for further evaluation. Clinical & experimental ophthalmology 37, no. 5 (2009): 514-523. Journal of Liposome Research 26, no. 4 (2016): 288-296. Drug safety 32, no. 1 (2009): 1-18. Pharmaceutics 12, no. 11 (2020): 1124. The journal of sexual medicine 8, no. 10 (2011): 2894-2903. Ophthalmologica 227, no. 2 (2012): 85-89. Nippon Ganka Gakkai Zasshi 101, no. 7 (1997): 551-557.
Although some of the references were already listed in our first version of the manuscript (Clinical & experimental ophthalmology 37, no. 5 (2009): 514-523; The journal of sexual medicine 8, no. 10 (2011): 2894-2903), we kindly acknowledge the reviewer’s suggestion and have incorporated most of them (Journal of Liposome Research 26, no. 4 (2016): 288-296; Drug safety 32, no. 1 (2009): 1-18; Pharmaceutics 12, no. 11 (2020): 1124; Ophthalmologica 227, no. 2 (2012): 85-89) in the appropriate sections of the revised manuscript.